# Transcriptomic and Cytogenetic Analysis of Oxaliplatin-Resistant Colorectal Adenocarcinoma HCT116 Cells to Identify Markers Associated with Platinum Resistance

**DOI:** 10.3390/ijms26188869

**Published:** 2025-09-11

**Authors:** Alisa Morshneva, Olga Gnedina, Ksenia Fedotova, Natalija Yartseva, Nikolay Aksenov, Maria Igotti

**Affiliations:** Institute of Cytology, Russian Academy of Sciences, 194064 St. Petersburg, Russia; olga.o.gnedina@gmail.com (O.G.); k1234390@gmail.com (K.F.); ya.ya-natm27951@yandex.ru (N.Y.); aksenovn@gmail.com (N.A.); marie.igotti@gmail.com (M.I.)

**Keywords:** tumor cells, colorectal cancer (CRC), drug resistance, oxaliplatin, aneuploidy, cytogenetic analysis, transcriptomic analysis

## Abstract

Platinum-based chemotherapy resistance remains a critical barrier in colorectal cancer (CRC) treatment. In this study, cytogenetic karyotyping was combined with transcriptomic profiling (RNA-seq) to elucidate resistance mechanisms by comparing two CRC cell lines: oxaliplatin-sensitive HCT116 and its resistant derivative HCT116 oxpl-R. Karyotyping unveiled tetraploidization and extensive genomic rearrangements in resistant cells, accompanied by prominent transcriptomic changes: 1807 differentially expressed genes (1216 upregulated and 519 downregulated). Pathway enrichment highlighted altered redox homeostasis and metabolic adaptation. Specifically, HCT116 oxpl-R cells exhibited elevated reactive oxygen species (ROS) production and enhanced energy metabolism. These findings establish a direct association between structural genomic alterations, transcriptional rewiring, and functional phenotypes in platinum resistance, providing a framework for targeting metabolic vulnerabilities in refractory CRC.

## 1. Introduction

Colorectal cancer (CRC) remains one of the leading causes of cancer-related mortality worldwide [1], with chemotherapy resistance posing a major challenge in treatment efficacy [2]. Oxaliplatin (Oxpl) is a third-generation platinum-based chemotherapeutic agent used as first-line chemotherapy for colorectal cancer. It prevents tumor cell proliferation by forming DNA–platinum adducts, which ultimately leads to the death of dividing cells. Platinum drugs (cisplatin, oxaliplatin, carboplatin, etc.) are widely used in anti-cancer therapy throughout the world; however, the development of resistance significantly limits their clinical success. Tumor recurrence, which is most often a consequence of the incomplete effectiveness of antitumor therapy approaches, is one of the most critical problems of modern clinical oncology.

The drug resistance of cancer cells can be either intrinsic, pre-existing at the start of treatment, as a result of genetic changes accumulated during carcinogenesis, or acquired as a result of treatment. Acquired resistance to chemotherapeutic drugs, in particular, oxaliplatin resistance, is a multifactorial process involving alterations in DNA repair, drug detoxification, apoptosis, cellular signaling, and many others [3]. However, the precise molecular mechanisms underlying resistance remain poorly understood.

Chromosomal rearrangements and changes in gene expression affecting proliferation, survival, and cell response to damaging effects may play an important role in the formation of resistance to chemotherapy [4,5]. In this regard, comparative analysis of karyotypes and transcriptomes of oxaliplatin-sensitive and oxaliplatin-resistant cell lines is of considerable interest for understanding the mechanisms of drug resistance.

The HCT116 cell line, derived from a human colorectal carcinoma, is frequently employed in CRC research due to its well-characterized genetic background. By comparing oxaliplatin-sensitive and oxaliplatin-resistant HCT116 cells, researchers can uncover key genomic and transcriptomic alterations associated with drug resistance. Karyotype analysis provides insights into chromosomal instability and large-scale genomic rearrangements, while transcriptomic profiling reveals differentially expressed genes and signaling pathways that may drive resistance.

This article explores the establishment of oxaliplatin-resistant HCT116 cells and presents a comparative analysis of their karyotypes and transcriptomes against their drug-sensitive parental HCT116 cells. By identifying genomic and gene expression changes, this study aims to shed light on potential biomarkers and therapeutic targets to overcome oxaliplatin resistance in colorectal cancer.

## 2. Results

### 2.1. Generation of Oxaliplatin-Resistant Cells Based on the Human Colorectal Cancer Cell Line HCT116

In the presented work, a line of oxaliplatin-resistant human colorectal cancer cells was obtained by repeated pulse exposure of parental HCT116 cells to a high concentration of oxaliplatin (Oxpl). For this purpose, cells at a confluent density (70–80%) were treated with 150 μM oxaliplatin for 6 h, after which the culture medium was changed to fresh medium without oxaliplatin. The concentration of 150 μM was selected based on MTT assay data as the half-maximal inhibitory concentration (IC50) of parental HCT116 cells at which 50% of the cells died 48 h after changing the medium to fresh without the active drug (IC50 48 h) (Figure 1a). In this experiment, the parental HCT116 cells were treated with different concentrations of Oxpl (50–250 μM) for 6 h, after which the culture medium was changed to a fresh one without oxaliplatin, and the cells were further cultured for another 48–144 h.

To develop resistance, the parental HCT116 cells were subjected to sequential rounds of oxaliplatin treatment. For this purpose, cells at a confluent density (70–80%) were treated with 150 μM oxaliplatin for 6 h, after which the culture medium was changed to fresh medium without oxaliplatin. The oxaliplatin concentration of 150 μM was chosen based on the results of the MTT test, as the concentration at which 50% of the cells died 48 h after changing the medium to fresh without the active drug (IC50 48 h) (Figure 1a).

After a recovery period and monolayer formation, the cells were subcultured twice. After repeated passages, the cells were subjected to the next round according to the method described above. The period of recovery of cell viability and their return to active proliferation was reduced with each subsequent round of oxaliplatin treatment. Thus, the duration of restoration of the active proliferation of cells that survived after the first round was 6–7 weeks, after the second round, it was 4–5 weeks, and after the third, it was 3–4 weeks.

After resumption of active proliferation, the cells were characterized by an increased size compared to the parental sensitive cells, with the presence of a large number of multinucleated cells.

After two rounds of Oxpl treatment, the cell resistance index (RI) was determined using the MTT test data, which is an important parameter for assessing the degree of their resistance to a chemotherapy drug or physical impact [6]. RI is calculated as the ratio of IC50 of resistant and sensitive cells (RI = IC50(HCT116 oxpl-R)/IC50(HCT116)). Cell resistance is considered high if RI > 10 and average if RI is in the range of 2–10. At RI in the range of 0–2, cells are considered sensitive to the effect [7,8]. According to the MTT test, the resistance index of the HCT116 oxpl-R cells after three rounds of oxaliplatin treatment was 7, and after four rounds, it was 10 (Figure 1b,c). Thus, it was shown that with an increasing number of rounds of treatment with 150 μM oxaliplatin for 6 h, the RI of HCT116 oxpl-R cells increases.

### 2.2. General Characteristics of the Obtained Cells Resistant to Platinum Drugs

The drug resistance of tumor cells can be either passive, due to a decrease in the intracellular concentration of the active substance, or active, which involves the restoration of damage and survival under conditions of cellular stress caused by the action of chemotherapeutic drugs [9]. In order to identify the type of resistance formed, an analysis of the distribution of the parental sensitive HCT116 cells and oxaliplatin-resistant HCT116 oxpl-R cells by the cell cycle was performed (Figure 1d). The concentration of 25 µM oxaliplatin was selected based on the dose–response curves (Figure 1b). This concentration induces a moderate toxic effect in sensitive cells while showing only a slight effect in resistant cells, thereby clearly illustrating the differential response between the two cell lines.

According to the results of the FACS analysis, oxaliplatin causes cell cycle arrest at the G2 checkpoint and the accumulation of sensitive HCT116 cells at the G2/M phase boundary, with subsequent initiation of the cell death program, as evidenced by an increase in the proportion of cells with sub-diploid DNA content (Figure 1d). Resistant HCT116 oxpl-R cells also accumulate in the G2/M phase under Oxpl treatment, but the cell cycle block is not accompanied by an increase in the proportion of cells with sub-diploid DNA content.

Thus, it was demonstrated that Oxpl causes cell cycle arrest, a characteristic of cells with damaged DNA, both in sensitive and resistant HCT116 cells, which is apparently associated with the accumulation of platinum DNA adducts under the action of Oxpl. This indicates that the formed resistance of HCT116 oxpl-R cells is not associated with tolerance mechanisms, which consist of avoiding damage and reducing the intracellular concentration of the active substance, but is realized through mechanisms of active resistance.

Drug-resistant tumor cells are often characterized by cross-resistance to other drugs. The width of the spectrum of this intersection can be used to judge the specificity of the resistance formed. To assess the degree of cross-resistance to other DNA-damaging drugs used at the IC50 concentration for the parental sensitive HCT116 cells, a comparative analysis of their effect on the viability of sensitive and oxaliplatin-resistant cells was carried out (Figure 1e). According to the MTT assay results, the resistant HCT116 oxpl-R cells exhibit a reduced response to other DNA-damaging drugs, namely, cisplatin and etoposide, compared to the parental cells. Resistance to doxorubicin was not affected.

### 2.3. Cytogenetic Analysis of Oxaliplatin-Resistant Human Colorectal Cancer Cells, HCT116 Oxpl-R, Revealed Multiple Karyotype Abnormalities

One of the most common features of cancer cells is an aneuploid karyotype. Approximately 90% of solid tumors and more than 50% of hematopoietic cancer cells exhibit varying degrees of aneuploidy [5]. Chromosomal instability has been associated with the acquisition of drug resistance [10,11,12]. Therefore, we compared the DNA content of the parental HCT116 cells and platinum-resistant HCT116 oxpl-R cells.

According to the flow cytometry data, the resistant HCT116 oxpl-R cells are characterized by an increased DNA content compared to the parental HCT116 cells (Figure 2a).

To analyze the changes in cell ploidy, as well as their karyotype during the development of resistance to oxaliplatin, a karyological analysis of the HCT116 oxpl-R cells was performed.

According to the cytogenetic analysis, the population of the HCT116 oxpl-R cells is hypotetraploid and heterogeneous in chromosome number, which varies from 80 to 88, mainly due to changes in the copy number of normal chromosomes ranging from 1 to 5.

Cells with chromosome numbers 84 and 85 constitute two modal classes in the population by chromosome number (Table 1, Figure 2c and Appendix A). The copy number varies for chromosomes 3, 5, 14, 15, 17, 18, 20, 21, and 22. The Y chromosome was detected in 7 of 28 karyotyped cells. The number of chromosomes was determined in 100 metaphase plates. To avoid random loss of chromosomes, cells with a certain number of chromosomes were taken into account if there were three or more cells with that number.

Six permanent clonal structural chromosome aberrations (SCAs) were identified, involving chromosomes X, 5, 6, 8, 10 (twice), 16, 17, and 18. The clonal SCAs include translocations t(17q;18q) (present in two copies), t(5q;6q), and t(8q;16p); additional material on the long arm of chromosome 10 (aad(10)); and the deletion of nearly the entire long arm of chromosome 10 (del(10)(q22)). Using spectral karyotyping [13], characterized aad(10) in parental HCT116 cells as a partial duplication of chromosome 10 (dup(10)). This dup(10) is present in two copies in all karyotyped HCT116 oxpl-R cells.

The parental HCT116 cells are pseudo-diploid with the chromosomal formula 46,XY,der(10)dup(10)(q22q23),der(8),t(8;16)(q13;p13.3),der(18)t(17;18)(q11.2;p11.2) (Appendix A). There are two chromosome translocations, namely, t(8;16) and t(17;18), and a duplication of chromosome 10—dup(10). The cell karyotype notation is given according to ISCN, the International System for Human Cytogenetic Nomenclature [14].

Comparison of the karyotypes of the HCT116 oxpl-R cells and the parental HCT116 cells revealed the following: (1) an increase in ploidy in the HCT116 oxpl-R cells to a hypotetraploid state and (2) the emergence of two new stable chromosomal aberrations in the HCT116 oxpl-R cells: a translocation t(?5q;6q) and a deletion of the long arm of chromosome 10 (del(10)(q22)). It can be hypothesized that the translocation t(?5;6)(q22;q25) may contribute to the acquisition of drug resistance in these cells. Additionally, a modification in one homologue of the X chromosome was observed in some cells. Notably, all HCT116 oxpl-R cells retained three chromosomal rearrangements characteristic of the original HCT116 line. These aberrations were present in two copies in the resistant cells, which may play a role in sustaining their proliferative capacity and contributing to genomic instability.

### 2.4. Transcriptomic Landscape of Oxaliplatin-Resistant HCT116 Cells Reveals Metabolic and Genome Stability Alterations

There is a chromosomal concept explaining drug resistance, which is based on the evidence that drug resistance is proportional to the number of chromosomal alterations in cells [4]. Obviously, changes in karyotype can alter the whole pattern of gene expression. Changes in some individual genes provide cells with an advantage during cytotoxic exposure, thereby driving drug resistance. To identify exactly how changes in the karyotype observed in the oxaliplatin-resistant cells are linked with alterations in gene transcription and to identify particular markers of resistance to oxaliplatin, transcriptomic analysis of the HCT116 oxpl-R and HCT116 cells was performed (Figure 3).

All six samples exhibited consistently high Pearson correlation coefficients with each other (ranging from 0.97 to 1.00), reflecting strong reproducibility across replicates (Appendix A). Correspondingly, principal component analysis (PCA) revealed clear clustering of the samples according to their biological groups, revealing a prominent difference in transcriptomic signatures between the sensitive and oxaliplatin-resistant cells (Figure 3). This concordance between correlation and clustering supports the absence of technical outliers. Consequently, the high degree of similarity among the samples confirms the robustness and quality of the dataset, justifying its use for downstream analyses.

Comparative analysis of gene expression between the oxaliplatin-sensitive HCT116 and oxaliplatin-resistant HCT116 oxpl-R cells identified 1807 differentially expressed genes (DEGs) using a false discovery rate (FDR) threshold of <0.05 and an absolute log2 fold change (|log2FC|) > 2. Among these, 1216 genes were significantly upregulated, while 591 genes were downregulated in the experimental group relative to the control (Figure 3b,c).

A heatmap was generated to visualize the expression patterns of the top 30 DEGs across the samples. The top DEGs were grouped into six distinct clusters based on their expression patterns across the samples, where each cluster represents a set of genes with similar expression dynamics.

Functional analysis was performed to determine whether there is enrichment of known biological functions, interactions, or pathways among the gene sets obtained. Gene set enrichment analysis (GSEA) was performed using the Gene Ontology (GO) and Kyoto Encyclopedia of Genes and Genomes (KEGG) databases.

According to the GO database, significant enrichment of upregulated genes was observed across a broad spectrum of biological processes, including respiration and energy metabolism (GO:0019646 aerobic electron transport chain, GO:0006754 ATP biosynthesis), GO:1902600 proton transmembrane transport, and GO:0044355 clearance of foreign intracellular DNA, among others (Figure 4a). In contrast, downregulated genes were predominantly enriched in pathways associated with chromosome organization and stability (GO:0033044, GO:0000819, GO:0051276, GO:0007059, GO:0098813), DNA repair (GO:0006281), and mitotic cell cycle process (GO:1903047).

The KEGG pathway analysis revealed activation of pathways controlling energy metabolism (hsa00190 oxidative phosphorylation) and protein synthesis (hsa03010 ribosome), ECM–receptor interaction (hsa04512), glycosphingolipid biosynthesis (hsa00601), and a set of disease and pathogen-driven pathways (hsa05208, hsa05171, hsa05016, hsa05218) (Figure 4b). Suppressed KEGG pathways are mostly related to genetic and epigenetic regulation and RNA processing: ATP-dependent chromatin remodeling (hsa03082), polycomb repressive complex (hsa03083), mRNA surveillance pathway (hsa03015), and spliceosome (hsa03040). In addition, there are also pathways regulating cell proliferation, nucleocytoplasmic transport (hsa03013), cell–cell interactions, and hormonal regulation (hsa04520, hsa04928).

These findings suggest that oxaliplatin-resistant cells exhibit reduced proliferative potential and increased genome instability compared to oxaliplatin-sensitive HCT116 cells. We suggest that the high level of oxidative stress in resistant cells may contribute to their reduced proliferative capacity and increased genomic instability. Excessive accumulation of reactive oxygen species (ROS) causes damage to DNA, proteins, and lipids, disrupting normal cell cycle mechanisms and leading to delays in cell division. Moreover, oxidative damage to the genome increases mutation frequency and chromosomal alterations, thereby promoting genomic instability and disturbing cellular homeostasis.

### 2.5. Oxaliplatin-Resistant Cells Exhibit Higher Redox Gene Activity and Enhanced Reactive Oxygen Species Production

The levels of reactive oxygen species and antioxidant system activity are often reported as indicators of malignancy and aggressiveness in tumor cells [15,16,17]. According to our transcriptomic data, oxaliplatin-resistant cells activate the expression of genes involved in redox homeostasis. For instance, the KEGG pathway associated with chemical carcinogenesis and reactive oxygen species (ROS), hsa05208, is generally upregulated in resistant cells (Figure 5a).

Our results revealed an upregulated expression of multiple redox-related genes in the oxaliplatin-resistant cells HCT116 oxpl-R (Figure 5b), supported by transcriptomic analysis and further validated through qPCR (Figure 5c).

In particular, increased expression levels were observed for genes involved in the antioxidant defense system, such as *GPX3*. However, transcription of glutathione reductase gene *GSR*, encoding a key enzyme in the glutathione antioxidant system, was suppressed in the resistant cells. Genes encoding members of the aldo-keto reductase family, AKR1C2 and AKR1C3, which are involved in detoxifying reactive oxygen species, were also upregulated. Activation of mitochondrial components was noted for genes encoding respiratory chain subunits—*NDUFB7*, *NDUFB5*, and *NDUFA2*. Additionally, elevated expression of *NCF2*, a gene involved in the formation of the NADPH oxidase complex, and *AHR*, a regulator of cellular response to xenobiotics, was observed. This pattern of activation indicates a complex reprogramming of redox balance and energy metabolism in the resistant cells.

To confirm the difference in redox status, ROS levels were examined in the HCT116 and HCT116 oxpl-R cells, measuring fluorescence intensity of DCF (Figure 5d,f) and mitochondrial membrane potential using TMRM staining (Figure 5e). Indeed, the oxaliplatin-resistant cells had higher ROS levels compared to the oxaliplatin-sensitive HCT116 oxpl-R cells, as evidenced by the shift in the DCF peak shown by flow cytometry (Figure 5d) and directly illustrated through fluorescent microscopy of DCF-stained cells (Figure 5f). At the same time, there was no shift in TMRM (Figure 5e), indicating that despite the elevated levels of ROS, the mitochondria maintain a normal membrane potential, meaning their energy function and ability to sustain the potential remain intact.

### 2.6. The Relationship Between Transcriptomic Changes and Chromosomal Rearrangements in Oxaliplatin-Resistant HCT116 Oxpl-R Cells

To associate these transcriptomic signatures with karyotype changes, the distribution of DEGs throughout chromosomes was visualized in the HCT116 oxpl-R cells on a karyoplot (Figure 6a). According to the cytogenetic data obtained, HCT116 oxpl-R are hypotetraploids relative to HCT116 cells. After tetraploidization, proportions of gene transcripts are supposed to stay the same; however, it is not just proportional scaling [18,19], so tetra- and octaploids are known to have changed gene expression patterns [20]. While these tetraploidization-induced transcriptomic changes are not directly evident in the karyoplot, the HCT116 oxpl-R genome shows multiple aneuploidies, including three copies of chromosomes 2, 4, 5, 6, 8, 13, 15, and 16 and five copies of chromosomes 3 and 22. These disproportionate karyotype changes are reflected in the transcriptomic data, with specific chromosomes showing enrichment of either downregulated or upregulated DEGs. In the HCT116 oxpl-R cells, chromosomes with three copies (chr2, 4, 13, 15, and 16) were predominantly enriched with downregulated DEGs, whereas the five-copy chromosome 3 was markedly enriched with activated genes, accounting for over 10% of all upregulated DEGs (Figure 6b). Additionally, downregulated DEGs are enriched at the distal end of the long arm of chromosome 10. The Y chromosome was lost in a large portion of the cell population, which explains that it carries only downregulated DEGs. It should be noted that this approach to interpreting the transcriptomic effects of chromosomal rearrangements is somewhat straightforward and simplistic, as it does not take into account possible upstream regulation of gene transcription.

Given the extensive and heterogeneous karyotypic alterations, where different cells within the population carry varying chromosome numbers, it is highly challenging to unequivocally interpret transcriptomic changes. What is detected as a significantly differentially expressed gene (DEG) may represent either a true change in transcript abundance within individual cells due to gene expression regulation or expression confined to only a subset of the cellular population. It remains unclear to what extent these clonal variations in gene expression contribute to the development of platinum resistance.

## 3. Discussion

Given that oxaliplatin stands as a cornerstone in the therapeutic management of colorectal cancer (CRC), frequently integrated into regimens such as FOLFOX [21], the resistance of colorectal cancer cells to oxaliplatin remains a significant clinical challenge, and the detection of resistance markers in colorectal cancer cells is still being studied using transcriptomic techniques. To date, a considerable number of transcriptomic studies have been conducted to identify molecular signatures associated with oxaliplatin resistance in CRC cells, underscoring the complex, multifactorial nature of chemoresistance.

Transcriptomic profiling of oxaliplatin-resistant colorectal cancer (CRC) cells derived from DLD1 and HCT116 identified ALCAM activation with concurrent CD22, CASP1, and CISH suppression as key molecular features of oxaliplatin non-response [22]. Five genes (COPE, P4HA1, ATF6, IBTK, and PHLDB3) were identified from DEGs associated with oxaliplatin resistance in three CRC cell lines (HCT116, HT-29, and LoVo). These genes showed a significant association with poor prognosis in colorectal cancer patients [23]. Oxaliplatin resistance development has been linked to increased miR-29a-3p expression, which leads to reduced levels of the ubiquitin ligase component FEM1B and elevated Gli1 levels. TCGA cohort analyses confirm that oxaliplatin resistance mediated by the miR-29a/FEM1B axis is clinically significant [24]. Analysis of the TCGA cohort of the tissue samples from patients with colorectal cancer and normal tissues revealed significantly elevated WBSCR22 expression in human CRC tissues compared to normal controls [25].

Despite the existing data, the task of elucidating mechanisms of resistance remains highly relevant, given that researchers studying the transcriptomes of drug-resistant cancer cells highlight the complexity of the problem, which stems from the alteration of entire gene clusters and signaling pathways instead of individual genes. In this regard, the discovery of universal markers or broadly applicable target panels for overcoming drug resistance in cancer cells seems unrealistic at present. Instead, priority should be given to the ongoing accumulation of this type of data, gathering new transcriptomic signatures from cells exhibiting drug resistance to enhance our insight into possible resistance mechanisms, which are often distinct for each resistant cell line.

The origin of the platinum resistance observed in HCT116 oxpl-R cells results from multiple cycles of high-concentration platinum drug exposure. Prolonged culturing itself is known to enable cells to develop resistance to treatment [26,27], and in this case, the cells went through 26–28 passages during the acquisition of oxaliplatin resistance. According to our results, oxaliplatin-resistant cells, HCT116 oxpl-R, demonstrate selective resistance to platinum drugs (cross-resistance) and only a slight cross-resistance to other DNA-damaging drugs (Figure 1e), supporting the conclusion that platinum exposure, rather than the duration of culturing, is the key factor driving resistance development in these cells.

What sets this study apart from others is that we analyzed the transcriptome of oxaliplatin-resistant HCT116 cells (HCT116 oxpl-R) alongside their karyotype, in an effort to link chromosomal alterations to the transcriptomic signatures observed. Such an integrative approach may provide deeper insights into the evolutionary pressures and genomic reorganizations accompanying the acquisition of resistance and contribute to improved prediction of potential resistance mechanisms through cytogenetic profiling of resistant cells.

Indeed, aneuploidy and chromosomal instability (CIN) are hallmarks of most solid tumors that contribute to tumor heterogeneity and selective adaptation [28]. High levels of aneuploidy in human tumors are often correlated with poor patient outcomes, supporting their role in certain aspects of tumor progression [5,29]. CIN has been reported to correlate with drug resistance, although a direct causal relationship has not been demonstrated [10,11]. Karyotypic heterogeneity within tumors generates genetically diverse subclones, facilitating adaptive selection of drug-resistant variants [29]. This intratumoral diversity, driven by chromosomal abnormalities, enables tumors to evolve resistance to multiple therapeutic agents. Chromosomal instability leads to frequent gains or losses of whole chromosomes or segments, affecting oncogenes or tumor suppressors and influencing cell fate under drug treatment. For example, Beroukhim et al. showed recurrent copy number alterations in cancers that can impact therapy response. These alterations include many known cancer target genes and oncogenes, such as members of the BCL2 family of apoptosis regulators and components of the NF-κB pathway [30].

The development of oxaliplatin resistance in CRC cells is a complex process accompanied by significant karyotypic alterations, as observed in our study. The resistant cell lines, derived from parental oxaliplatin-sensitive HCT116 cells, exhibited pronounced polyploidization and chromosomal rearrangements, notably the deletion of the long arm of chromosome 10 (10q) and the translocation between chromosomes 5 and 6 at bands 22q and 25q, denoted as t(5;6)(22q;25q). Deletion of the long arm of chromosome 10 has been identified in various cancers and has been associated with tumor development, progression, and, in some cases, drug resistance. Of a cohort of 260 patients with primary neuroblastoma tumors at disease onset, 26 exhibited 10q loss, while the remaining 234 displayed different segmental chromosomal aberrations (SCAs) [31]. In A-375 melanoma cells, Mps1 inhibition-induced SCRs caused chromosome 10 loss (observed in 8/11 cells) and chromosome 18 gain (8/11 cells). Single-cell analysis linked chr10 loss to paclitaxel resistance [32].

According to our data, oxaliplatin-resistant HCT116 cells undergo polyploidization (predominantly tetraploidization) in the process of acquiring platinum resistance. Tetraploidization, or whole-genome duplication, is increasingly recognized as a pivotal event in cancer drug resistance development. Polyploidy likely contributes to oxaliplatin resistance by providing redundant genetic material that mitigates the impact of DNA damage induced by oxaliplatin, a platinum-based agent that forms DNA crosslinks. This adaptation may stabilize the cancer genome against genotoxic stress while facilitating further chromosomal rearrangements. Studies have shown that tetraploid cancer cells, which contain double the normal chromosome number, are often more tolerant to chemotherapeutic agents and radiotherapy compared to diploid cells [33]. Experimental evidence in CRC lines (HCT116 and RKO) confirmed the intrinsic resistance of tetraploid cells against DNA damage-induced apoptosis [34]. Oliver and colleagues demonstrated that prolonged cisplatin treatment promotes the emergence of resistant tumors that possess an increased frequency of genomic alterations. Around 83% of cisplatin-resistant lung tumor cells harbored whole-chromosome aberrations, including gains and losses of entire chromosomes [35].

Apart from whole-genome polyploidization in HCT116 oxpl-R cells, aneuploidies affecting specific chromosomes should also be taken into account, as we observe complex polyploidy with chromosome-specific copy number variations that undoubtedly affect gene dosage in the transcriptome of oxaliplatin-resistant cells. For instance, trisomy of chromosomes 7 and 13 in DLD1 colorectal cancer cells has been reported to confer resistance to 5-fluorouracil [36]. In the HCT116 oxpl-R karyotype, there is a mix of 13 tetrasomic (1, 7, 9, 10, 11, 12, 14–20), 7 trisomic (2, 4–6, 8, 13, 21), and 2 pentasomic (3, 22) chromosomes (Figure 2 and Appendix A). Of special interest are chromosomes 3 and 22, which are pentasomic and account for the highest proportion of gene expression in the transcriptome. Chromosome 3 amplification is one of the most prevalent and significant alterations in lung cancer, contributing to its invasiveness and response to therapy [6]. Moreover, it has been reported that chromosome 3 mediates growth arrest and suppression of apoptosis [37]. Less is known in the literature about the association between chromosome 22 and tumor cell phenotypes. However, one study reported an over-representation of specific regions of chromosome 22 in human glioma cells linked to resistance [38]. Similar chromosomal copy number aberrations were observed in trastuzumab-resistant JIMT-1 cells, including 16 amplifications and 11 deletions. Genes in amplified regions showed overexpression, while those in deleted regions were underexpressed compared to diploid regions. [20].

The loss of the Y chromosome in resistant cells resulted in the downregulation of genes located on the Y chromosome, which appeared among the differentially expressed genes in our data, with suppressed expression in these resistant cells. The loss of the Y chromosome is widely documented in the literature as a negative prognostic marker and a characteristic feature of drug-resistant cancer cells [39,40].

These findings align with the recent literature on genomic instability and its role in chemotherapeutic resistance, providing insights into the molecular mechanisms driving resistance and prompting discussion about the interplay between karyotypic changes and selective pressures in cancer evolution. The overall increase in chromosome copy number leads to global gene expression shifts that support resistance phenotypes, such as improved DNA repair, evasion of apoptosis, and enhanced stress response.

The transcriptomic profile of resistant cells is intricately linked to karyotypic rearrangements, as these structural changes disrupt gene regulation and signaling cascades. Integrating transcriptomic data, our RNA-seq analysis of resistant cells highlighted extensive differential gene expression primarily affecting pathways related to energy metabolism and reactive oxygen species (ROS) homeostasis, such as the regulation of proton transmembrane transport (*SLC9A9*, *SLC36A1*, *SLC25A27*, *SLC25A20*, *UCP3*), pathway regulating ROS homeostasis and redox status (*NCF2*, *AKR1C1*, *AKT3*), aerobic electron transport chain and oxidative phosphorylation (*NDUFB5*, *NDUFA4L2*, *NDUFB7*), and ATP biosynthetic process (*DMAC2L*, *NDUFB5*, *PINK1*) (Figure 4). However, interpreting transcriptomic data in the context of karyotypic changes poses challenges. Each rearrangement triggers a cascade of downstream signaling alterations, creating a complex signaling context that is difficult to attribute to specific chromosomal events. For instance, translocation t(5;6)(22q;25q), harboring many of the solute carrier (SLC) family genes, may cause an altered expression profile of these genes. Thus, while transcriptomic data provide a snapshot of the resistant phenotype, disentangling the precise contributions of individual rearrangements remains a significant limitation, emphasizing the need for integrative multi-omics approaches.

Among our key DEGs, there are many genes involved in antioxidant defense and redox homeostasis pathways that modulate intracellular ROS levels (Figure 5). ROS play a dual role in resistance: while high ROS levels can induce cell death, resistant cells often upregulate antioxidant defenses to counteract oxidative stress induced by oxaliplatin [16]. The question of whether ROS are the cause or the consequence of drug resistance in cancer cells has a dual answer. A body of evidence indicates that ROS are important for both inducing and maintaining drug resistance. ROS can induce drug resistance by promoting genomic instability through DNA damage [41] and activating pro-survival signals that block apoptosis [15,42]. Conversely, maintaining the resistant phenotype requires a critical balance between ROS and antioxidant enzymes, highlighting ROS modulation as a potential therapeutic approach [43,44]. Interestingly, we identified several novel DEGs, such as AHR and AKR1C2/3, which have not been widely reported in the context of oxaliplatin resistance. These genes, involved in pathways that converge on ROS modulation, may represent novel therapeutic targets. The AHR protein is a key regulator of gene expression, driving detoxification, cellular homeostasis, and immune responses [45]. AHR modulates redox balance by transcriptionally upregulating antioxidant enzymes like NQO1 and GSTs through the Nrf2 pathway while simultaneously inducing prooxidant cytochrome P450 enzymes (CYP1A1/1B1) that can generate ROS during xenobiotic metabolism [46,47]. In the current literature, AHR is primarily implicated as a contributing factor to the development of resistance against targeted therapeutic agents [48,49]. Regarding aldo-keto reductases, AKR1C2 and AKR1C3 regulate intracellular ROS through complementary mechanisms. Both NADPH-dependent enzymes detoxify reactive carbonyls but target different substrates. AKR1C2 reduces lipid peroxidation products, preventing oxidative damage, while AKR1C3 modulates glutathione metabolism and PI3K/AKT signaling [46,50]. AKR1C3 also blocks ferroptosis by stabilizing HSPA5/GRP78 to maintain GPX4 levels. In chemoresistant cancers, their co-overexpression creates a robust antioxidant defense—AKR1C2 clears toxic aldehydes, while AKR1C3 sustains redox balance through GSH and survival pathways. Targeting these enzymes elevates ROS and restores drug sensitivity, offering therapeutic potential against resistant tumors. It was shown that AKR1C2 is upregulated in cisplatin-resistant cell lines of lung, cervical, and ovarian cancers, and its knockdown restores chemosensitivity [51,52]. In colon cancer, AKR1C3 downregulation by ARID3A enhances 5-FU sensitivity [53]. It was shown that AKR1C3 promotes resistance to (1) anthracyclines via carbonyl reductase activity [54,55], (2) paclitaxel in breast cancer via aldehyde metabolism [56], (3) docetaxel in prostate cancer [57], and (4) esophageal adenocarcinoma therapies through ROS/AKT regulation [50]. AKR1C3 inhibition may overcome chemoresistance across cancers by blocking drug detoxification and ROS modulation. The prominence of ROS-related pathways in our data underscores the importance of oxidative stress in shaping resistance, suggesting that targeting antioxidant defenses could sensitize resistant cells to therapy.

Some of the detected ROS-related genes are reported as prognostic markers for CRC, according to the Human Protein Atlas [58,59]. Changes in gene expression observed in oxaliplatin-resistant HCT116 oxpl-R cells have been associated with CRC patient survival outcomes. Notably, the suppression of the antioxidant gene GPX3 and the upregulation of NCF2 are both associated with poor prognosis (Appendix A). In contrast, *NDUFB5*, the gene encoding a subunit of NADH dehydrogenase (ubiquinone) and upregulated in resistant cells, is reported to be linked with positive prognosis in CRC according to the survival analysis data. However, active mitochondrial metabolism and biogenesis are reported to be associated with resistant CRC phenotypes [60], and the possible role of NDUF gene expression in CRC progression is still to be established. Overexpression of *NDUFA4L2*, which is also upregulated in HCT116 oxpl-R cells (log2 fold change = 2.14), is reported to be linked with poor prognosis in colon cancer [61] and serves as a prognostic biomarker for most cancers, including colon cancer [62].

In this study, an integrated analysis of the karyotype and transcriptome of oxaliplatin-resistant cells was conducted, uncovering numerous chromosomal rearrangements and a substantial set of differentially expressed genes that impact diverse biological pathways. While certain chromosomal and gene expression signatures align with previously described hallmarks of platinum resistance, others constitute novel findings that broaden the current understanding of resistance biology. A critical challenge is to discern which molecular and chromosomal changes are common hallmarks of platinum resistance versus those that stem from the unique evolutionary trajectory of our resistant cell lines. Addressing this, we aim to expand our research by screening a diverse collection of cell lines resistant to various platinum drugs and other therapies to extract robust, recurring patterns of genomic instability. Combined with the literature evidence, our study will facilitate the identification of new molecular markers for detecting tumor genomic instability and support the design of innovative therapies targeting drug resistance in tumor cells.

## 4. Materials and Methods

### 4.1. Cell Line

Human colorectal carcinoma cell line HCT116 was obtained (Praymbiomed, Moscow, Russia) and tested according to short tandem repeat (STR) standards by Gordiz (Moscow, Russia). Oxaliplatin-resistant HCT116 oxpl-R cells were obtained by repeated pulse exposure of parental HCT116 cells to a high concentration of oxaliplatin (Section 2.1) and had undergone 26–28 passages at the time of cryopreservation. The parental HCT116 and oxaliplatin-resistant HCT116 oxpl-R cells were cultured in DMEM supplemented with 10% fetal bovine serum and gentamicin at 37 °C and 5% CO_2_. The cell lines contain a Tet-On inducible genetic modification system. Without doxycycline, the transgene expression is off. We confirmed the lack of basal expression of the transgene prior to experiments, ensuring this modification did not affect baseline cell behavior.

### 4.2. Cell Viability Analysis

Cell viability was assessed using the MTT assay. For this, the cells were seeded into 96-well plates at a density of 15 × 10^3^ cells per well. Twenty-four hours after seeding, the cells were treated with the appropriate drugs for 48 h. Viability was determined spectrophotometrically by assessing the metabolic activity of the cells by their ability to reduce 3-(4,5-dimethylthiazol-2-yl)-2,5-diphenyl-tetrazolium bromide (MTT) (Sigma–Aldrich, St. Louis, MO, USA) to insoluble formazan. For this, the cells were incubated in an MTT solution in PBS, at a final concentration of 0.5 mg/mL, for 1.5 h at 37 °C in a CO2 incubator. Then, the culture medium was removed, and the cells were lysed in DMSO. Optical density was determined at 570 nm using a Multiscan-EX instrument (Thermo Labsystems, Shanghai, China) using DMSO as a control.

### 4.3. Distribution of Cells by DNA Content

The cells were washed with PBS and permeabilized with 0.01% saponin/PBS solution for 15 min at room temperature. The cells were then incubated in the presence of 100 μg/mL RNase A and 10 μg/mL propidium iodide for 15 min at 37 °C and analyzed on a CytoFLEX flow cytometer (Beckman Coulter, Breya, CA, USA).

### 4.4. Measurement of ROS Levels

For the detection of reactive oxygen species (ROS), the cells were incubated with 2′,7′-Dichlorodihydrofluorescein diacetate (DCFH-DA, D6883, Sigma), which is acetylated inside the cells to form a non-fluorescent compound that is subsequently oxidized by ROS to fluorescent 2′,7′-dichlorofluorescin (DCF). The cells were washed with warm PBS and incubated for 20 min at 37 °C in 5 µM DCFH-DA/PBS, washed again in warm PBS, and then analyzed using a flow cytometer Coulter Epics XL (Bechman, Brea, CA, USA) with an excitation wavelength of 488 nm.

### 4.5. Measurement of Mitochondrial Membrane Potential (Ψm)

The cells were incubated with 50 nM tetramethylrhodamine methyl ester (TMRM) in PBS for 10 min at 37 °C. TMRM, a cationic fluorescent dye, accumulates in mitochondria in a membrane potential (ΔΨm)-dependent manner. After staining, the cells were immediately analyzed using a CytoFLEX flow cytometer (Beckman Coulter, Brea, CA, USA) with 488 nm excitation and emission detection at 560–606 nm.

### 4.6. Karyotype Analysis (GTG-Banding)

Karyotyping of the human colorectal carcinoma cell line HCT116 and oxaliplatin-resistant HCT116 was performed. To accumulate cells in the metaphase stage, colchicine solution was added to the culture medium at a final concentration of 0.375 µg/mL, followed by a 1.5 h incubation. Then, the cells were removed with a trypsin–Versene solution, treated with a hypotonic solution (0.55% KCl and 1% Na citrate solutions) for 20 min at 37 °C, and fixed using a 3:1 methanol–glacial acetic acid solution for 15 min. The fixing solution was changed to a fresh one. The procedure was repeated 3 times. For metaphase chromosome spread preparation, the fixed cell suspension was dropped onto dry slides over a 50 °C water bath and then left to dry completely. The dried chromosome preparations were treated with 0.02% trypsin solution for 3 min at 30 °C, and the enzymatic reaction was halted with blocking solution (134 mM KCl; 3,42 M NaCl; 104 mM NaHCO3; 140 mM a-D-glucose) for 15 s. The chromosomes were stained with 2% Giemsa solution (Paneco, Moscow, Russia) for 4 min. Chromosomal structural rearrangements were automatically analyzed using a hardware–software system comprising an AxioSkop A1 ProgRes MF microscope (Carl Zeiss Microscopy, Jena, Germany), a high-resolution monochrome CCD camera (Jenoptik, 1360 × 1024 pixels, Jena, Germany), and a computer running VideoTesT Karyo 3.1 software.

### 4.7. RNA Sequencing and Transcriptomic Data Analysis

RNA Isolation: For transcriptome analysis, total RNA was extracted from the HCT116 and HCT116 oxpl-R cells using the Lira reagent (Biolabmix, Novosibirsk, Russia) according to the manufacturer’s protocol.

For RNA quality assessment, the concentration of extracted nucleic acid was detected with Nanodrop2000 (manufacturer: ThermoFisher, South San Francisco, CA, USA, model: Nanodrop2000), and the integrity was detected with Agient2100, LabChip GX (manufacturer and model: Perkin Elmer LabChip GX). A Hieff NGS Ultima Dual-mode mRNA Library Prep Kit for Illumina (Yeasen Tec. Inc., Shanghai, China), model:13533ES96, was used for library construction. Product purification magnetic beads, namely, HieffNGS DNA selection Beads (Superior Ampure XP alternative) (Yeasen), model:12601ES56, were used. The primer adapters included the following: adpter3 = “AGATCGGAAGAGCACACGTCTG AACTCCAGTCAC” and adpter5 = “AGATCGGAAGAGCGTCGTGTAGGGAAAGAGTGT”. The libraries were sequenced on the Illumina Novaseq X platform (Illumina, San Diego, CA, USA). The PE150 sequencing reagent was NovaSeq X S4 Reagent Kit (San Diego, CA, USA).

The average number of pair-end reads was 40.1 ± 3.5 million per sample in this study. After quality filtering and trimming with Trimmomatic [63] (HEADCROP:10 SLIDINGWINDOW:4:15 LEADING:3 TRAILING:3 MINLEN:36), the average survival rate for the reads was 98%.

Transcript quantification was performed via lightweight mapping using Salmon [64] (version 1.6.0). Mapping was conducted in selective alignment mode through a two-stage process: reference index construction and read mapping. Decoy sequences representing genomic regions with high sequence similarity to annotated transcripts were generated based on the human reference genome GRCh38 Gencode (release 47). A “decoy-aware” reference index was built based on a gentrome (concatenated genome and transcriptome sequences from GRCh38 Gencode release 47). Read mapping was performed with additional arguments: —validateMappings—numBootstraps 30—seqBias—gcBias.

The transcript-level quantifications obtained with Salmon were imported and summarized to the gene level using the tximport R package (version 1.34.0) [65]. Genes with zero counts across all samples were filtered out prior to analysis. Differential gene expression analysis was performed in R using the DESeq2 [66] package (version 1.46.0). Raw count data, generated from transcript quantification, were used for analysis. The workflow included normalization for sequencing depth, estimation of dispersion parameters, and fitting of a negative binomial generalized linear model. A variance stabilizing transformation (VST) was performed for downstream visualization and quality assessment. Differential expression testing was conducted using the Wald test with an adjusted *p*-value cutoff (Benjamini–Hochberg correction) of 0.05 to identify significantly differentially expressed genes.

To plot DEGs according to their chromosome location, the karyoploteR package for R was used [67].

Functional enrichment analysis was performed using the R package clusterProfiler (version 4.14.6) [68] in conjunction with the org.Hs.eg.db annotation database for Homo sapiens. DESeq2 Wald test statistics were extracted and sorted to create a gene ranking suitable for gene set enrichment analysis (GSEA) [69]. GSEA was conducted against the KEGG (Kyoto Encyclopedia of Genes and Genomes) and GO (Gene Ontology) pathway databases using the gseKEGG and gseGO functions, respectively, with a significance cutoff of adjusted *p*-value < 0.05 (Benjamini–Hochberg correction). The GseGO function was used with the following parameters: minGSSize = 3, maxGSSize = 800, pvalueCutoff = 0.05, and OrgDb = org.Hs.eg.db. The results obtained were simplified using the “simplify” method from the package clusterProfiler (cutoff = 0.7, by = “p.adjust”, select_fun = min) [68] to reduce redundancy among enriched Gene Ontology (GO) terms. The GseKEGG function was used with default parameters. The resulting enriched pathways were converted to a readable format and visualized using dot plots with the enrichplot package (version 1.26.6) [70], facilitating interpretation of functional patterns associated with differential expression.

### 4.8. Real-Time PCR Analysis

Reverse transcription was performed with 2 µg of RNA using an RT kit (Biolabmix, Novosibirsk, Russia). The PCR program was as follows: 38 cycles of denaturation at 95 °C for 30 s, annealing at 60 °C for 30 s, and elongation at 72 °C for 30 s. Gene expression levels were normalized to *GAPDH*, and relative quantification was performed using the 2^^−ΔΔCt^ method (Life technologies), as described by Livak and Schmittgen [71].

All oligonucleotides used for qPCR are listed in Table 2 (F = forward primer, R = reverse primer). The primers were designed to cover conserved regions across all coding transcript variants of each gene, rather than a single specific splice variant, to ensure that the PCR product was independent of alternative splicing events.

## 5. Conclusions

Through combined analysis of chromosomal structure and gene expression, we uncovered significant genomic rearrangements and diverse transcriptomic shifts in oxaliplatin-resistant cells. While some of these changes align with established platinum resistance mechanisms, others reveal novel pathways and potential targets. Together with the existing literature, our results provide a foundation for identifying new markers of tumor genomic instability and contribute toward the development of targeted therapeutic approaches against chemoresistant tumors.

## Figures and Tables

**Figure 1 ijms-26-08869-f001:**
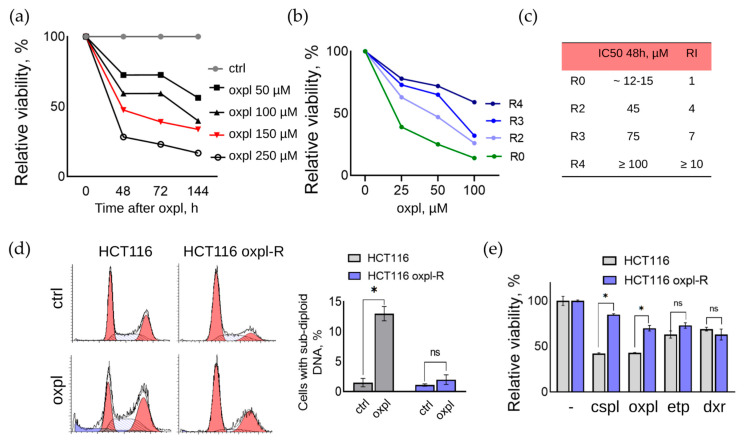
Generation and characterization of Oxpl-resistant cells: (**a**) MTT assay results. Parental HCT116 cells were treated with oxaliplatin at different concentrations for 6 h. Cell viability was determined 48−144 h after treatment with Oxpl. Results are presented relative to the viability of untreated control cells, which was set to 100%. (**b**) MTT assay results. Changes in HCT116 cell viability (R0) after 2−4 rounds of Oxpl treatment (R2, R3, R4) as resistance develops. Cells were treated with increasing concentrations of Oxpl for 48 h. The results are presented relative to the viability of untreated cells, which was taken as 100%. (**c**) Changes in the resistance index of HCT116 cells after rounds of Oxpl treatment. (**d**) Analysis of cell cycle distribution of parental sensitive HCT116 cells and resistant HCT116 oxpl-R cells exposed to 25 μM oxaliplatin for 48 h. Graphs of cell cycle distribution (cells with sub-diploid DNA content is indicated in blue on the graph) and a diagram of the proportion of cells with sub-diploid DNA content in percent. Error bars are plotted based on the standard error of the mean (SEM). The Mann–Whitney test was used to test the significance of differences (*—*p* < 0.05, ns—non-significant). (**e**) MTT assay results. Change in viability of oxaliplatin-resistant versus sensitive HCT116 cells exposed to chemotherapeutic agents (10 μM oxaliplatin, 25 μM cisplatin, 30 μM etoposide, and 0.5 μg/mL doxorubicin) after 48 h of exposure. The graph represents the average of 3 independent experiments; the error bars are based on the standard error of the mean (SEM). The Mann–Whitney test was used to test the significance of differences (* *p* < 0.05).

**Figure 2 ijms-26-08869-f002:**
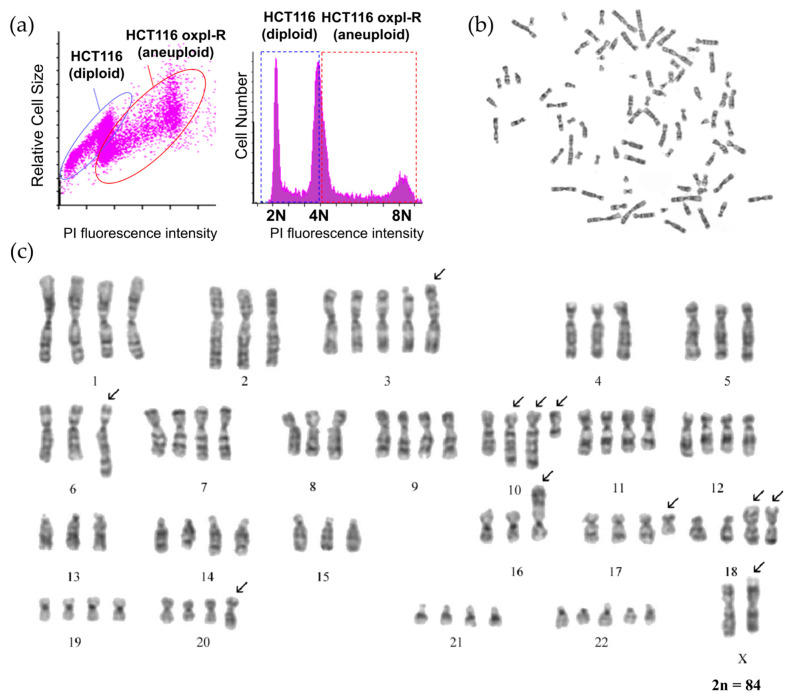
Cytogenetic characteristics of Oxpl-resistant HCT116 cells: (**a**) Distribution of HCT116 and HCT116 oxpl-R cell mixture by DNA content (flow cytometry), (**b**) metaphase plate, and (**c**) karyogram of an HCT116 oxpl-R cell with 84 chromosomes. Numbers 1–22 correspond to autosomes (chromosomes 1–22), while X corresponds to chromosome X. The arrows indicate the chromosome rearrangements.

**Figure 3 ijms-26-08869-f003:**
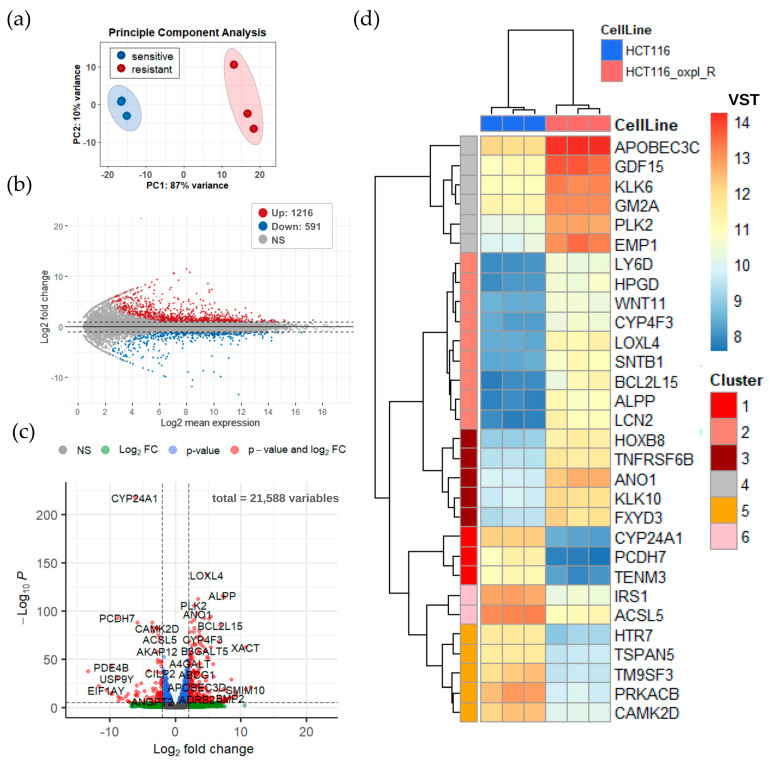
Differentially expressed genes (DEGs) in HCT116 oxpl-R vs. HCT116 cells. (**a**) Principal component analysis (PCA) based on the expression profile of HCT116 (red) and HCT116 oxpl-R (blue) samples. (**b**) MA or mean-difference plot (log2 fold change vs. log2 mean expression), fdr cutoff = 0.05, fc cutoff = 2. (**c**) Volcano plot with main DEGs (log2 fold change cutoff = 2, *p*-value cutoff = 10 × 10^−6^). (**d**) Heatmap with top 30 DEGs. Each row in the heatmap represents a gene, while each column corresponds to a sample. The color intensity reflects the normalized and variance-stabilized expression values (VST), with higher expression levels indicated by warmer colors and lower expression levels by cooler colors. Hierarchical clustering was applied to both genes and samples, allowing for the identification of groups with similar expression profiles.

**Figure 4 ijms-26-08869-f004:**
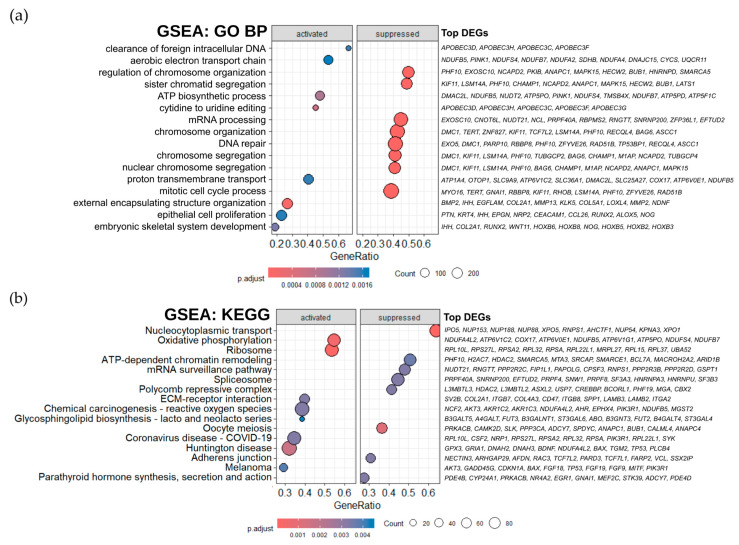
Functional analysis of transcriptomic changes in HCT116 oxpl-R vs. HCT116. Gene set enrichment analysis based on the GO (**a**) and KEGG (**b**) databases. The scale represents BH (Benjamini–Hochberg) adjusted *p*-values (pvalueCutoff = 0.05). Top DEGs were selected based on log2 fold change and are presented in descending order.

**Figure 5 ijms-26-08869-f005:**
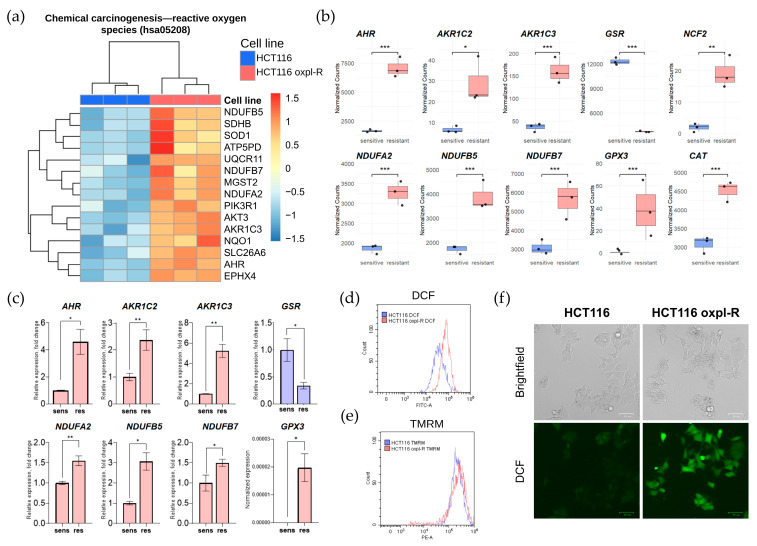
Redox homeostasis in HCT116 oxpl-R vs. HCT116 cells. (**a**) Heatmap with the top 15 DEGs from the KEGG pathway associated with chemical carcinogenesis and reactive oxygen species (ROS): hsa05208. The color intensity reflects the normalized and row-scaled variance-stabilized expression values (VST). (**b**) The boxplots display the distribution of normalized counts for selected redox-related genes in HCT116 (sensitive) and HCT116 oxpl-R (resistant) cells, with each point representing a normalized count for an individual sample. The significance shown on the plots is based on the adjusted *p*-values (p.adj) calculated using the Wald test (DESeq2). (**c**) Dynamics of redox-related gene expression revealed with qPCR. Normalized expression of redox-related genes *AHR, AKR1C2*, *AKR1C3*, *GSR*, *NDUFA2*, *NDUFB5*, *NDUFB7,* and *GPX3* in HCT116 and HCT116 oxpl-R cells, relative to HCT116. Expression of GPX3 in sensitive cells was not detected; therefore, for this gene, the graph is plotted using the absolute values of expression rather than fold change. The color of each bar denotes whether a gene is upregulated in sensitive cells (blue) or in resistant cells (red) compared to the other cell type. Expression of the GAPDH gene serves as the endogenous control. Data in (**b**,**c**) represent biological triplicate experiments and are displayed as mean ± SEM. The Mann–Whitney test was used to test the significance of differences (***—*p* < 0.001, **—*p* < 0.01, *—*p* < 0.05). (**d**,**e**) ROS levels (DCF fluorescence) (**d**) and mitochondrial membrane potential (TMRM fluorescence) (**e**) in HCT116 (blue) and HCT116 oxpl-R (red) cells according to flow cytometry data. (**f**) Representative fluorescent images of DCF-stained HCT116 and HCT116 oxpl-R cells.

**Figure 6 ijms-26-08869-f006:**
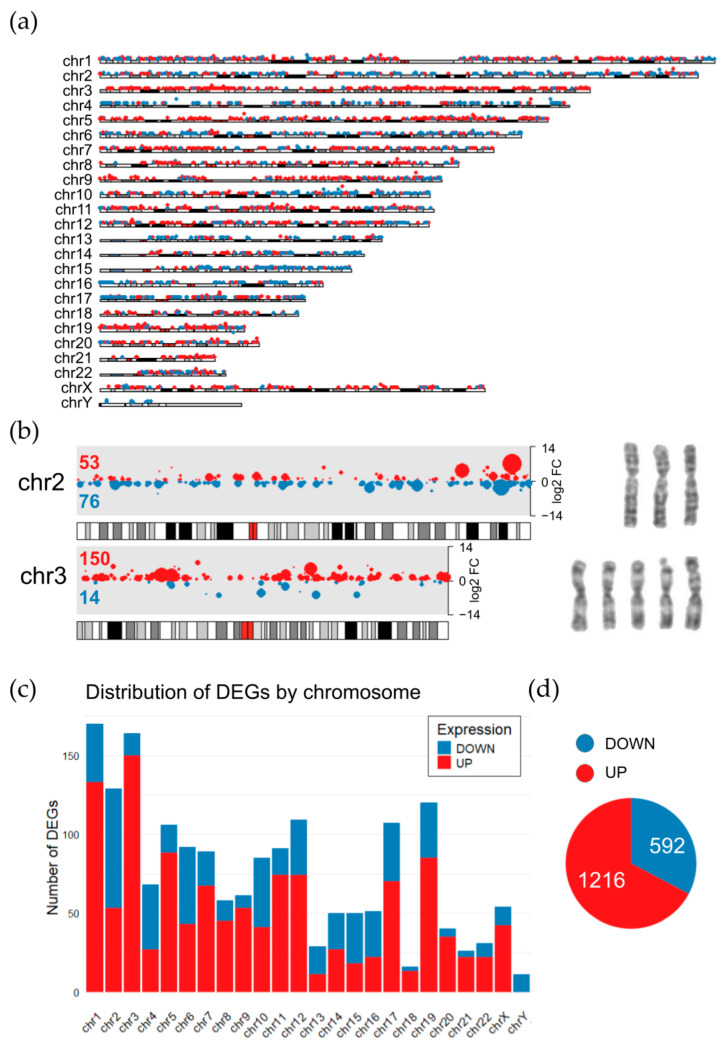
Distribution of DEGs by genome in HCT116 oxpl-R cells. (**a**) Karyoplot of upregulated (red) and downregulated (blue) genes according to their chromosome region. (**b**) Selective karyograms and karyoplots of aneuploid chromosomes (chr2 and chr3), numbers on plots represent the number of upregulated (red) and downregulated (blue) genes for each chromosome. In karyoplots (**a**,**b**), data is plotted opposite the ideograms of cytobands, meaning that the plotted data is visually aligned or positioned alongside the ideogram representation of chromosomes that show banding patterns (cytobands). Point sizes are inversely proportional to the adjusted *p*-value (i.e., smaller *p*-values correspond to larger points), scaled by the transformation for optimal visualization. (**c**) Barplot with the number of DEGs on each chromosome. (**d**) Pieplot representing the total number of up- and downregulated genes.

**Table 1 ijms-26-08869-t001:** Distribution of chromosomes by number in HCT116 oxpl-R cells.

N Chromosomes	80	81	82	83	84	85	86	87	88
N cells	3	4	6	10	23	25	11	5	6

**Table 2 ijms-26-08869-t002:** Oligonucleotides used for qPCR.

Gene ID	Gene	Forward (F)	Reverse (R)
2597	*GAPDH*	ACCATCTTCCAGGAGCGAGA	GACTCCACGACGTACTCAGC
2936	*GSR*	AACGAGCTTTACCCCGATGT	GGATCCCAACCACCTTTTCT
2878	*GPX3*	GATCCTTCCTACCCTCAAGTATGT	TCCGAGGTGGGAGGACA
196	*AHR*	GGCCTGAACTTACAAGAAGGA	TGTATGACATCAGACTGCTGAA
4695	*NDUFA2*	CTGGGCTCTGCTGGACTTAG	CTTTGCCTCAGTGGTTGCAC
4711	*NDUFB5*	CCTAAGATTCAGGCCAGTGGG	GAATGAATGGCCTGTGGCTTT
4713	*NDUFB7*	GAGCACCGCGACTATGTGA	TGACTGGTCCATAGGGTGGG
1646	*AKR1C2*	ATGCGCCTGCAGAGGTTC	ACCTGCTCCTCATTATTGTAAACAT
8644	*AKR1C3*	GCTGGGTTCCGCCATATAGATTC	GGACCAACTCTGGTCGATGAAA

## Data Availability

The RNA-seq data reported in this article were deposited in NCBI’s Gene Expression Omnibus (GEO) and are accessible through GEO series accession number GSE304295.

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
