# Peer review of "Transcriptomic and Cytogenetic Analysis of Oxaliplatin-Resistant Colorectal Adenocarcinoma HCT116 Cells to Identify Markers Associated with Platinum Resistance"

_ijms, 2025, doi:10.3390/ijms26188869_

Round 1
Reviewer 1 Report
Comments and Suggestions for Authors
In this article the authors have investigated transcriptomic and karyological differences in oxaliplatin resistant colorectal cell line HCT116 relative to sensitive form. Resistance to oxaliplatin and its investigation has been documented earlier in many cancers (10.3390/cancers13040724). The rationale is and data is presented clearly. The data presented in this article is within the scope of the journal. Similar modeling for dysregulated genes, including chromatin alteration, has been investigated earlier by other groups (10.1038/s41419-025-07855-y; 10.1002/EXP.20220136); here the authors are presenting information on metabolic pathways and gene dataset. While not particularly novel, the data presented in this manuscript adds further information regarding resistant pathways and their DEG analysis. The authors can present umap analysis on several of the datasets they are showing and it would add in the favor of the manuscript’s quality. If the authors can demonstrate any information from patient tissues and compare it with the cell line data that would bolster the conclusion.
Author Response
Thank you very much for your attention to our study and for your constructive comments. Below, we will address your questions point by point:
1 “The authors can present umap analysis on several of the datasets”
- We appreciate the reviewer’s suggestion to include UMAP analysis. However, since our study is not based on single-cell data but on bulk samples with only two experimental points, we believe that PCA provides a sufficient overview of the data structure. UMAP is particularly useful for resolving fine-grained clusters in complex, high-dimensional single-cell datasets, which is not applicable in our case. Therefore, performing UMAP would not yield additional biologically meaningful insights beyond what PCA already conveys for our dataset. We believe the current analyses adequately address the study’s goals.
2 “If the authors can demonstrate any information from patient tissues and compare it with the cell line data that would bolster the conclusion.”
We are grateful to the reviewer for this helpful suggestion. We used the Human Genome Atlas and literature data to provide additional information regarding the prognostic value of reported targets in CRC. These findings are presented on the new Figure S4 and discussed in the text. Line 537: “Some of the detected ROS-related genes are reported as prognostic markers for CRC according to the Human protein atlas [52,53]. Changes in gene expression observed in oxaliplatin-resistant HCT116 oxpl-R cells have been associated with CRC patient survival outcomes. Notably, the suppression of the antioxidant gene GPX3 and the upregulation of NCF2 are both associated with poor prognosis (Figure S4).. In contrast, NDUFB5, the gene encoding a subunit of NADH dehydrogenase (ubiquinone) and upregulated in resistant cells, is reported to be linked with positive prognosis in CRC according to the survival analysis data. However, active mitochondrial metabolism and biogenesis is reported to be associated with resistant CRC phenotypes [54] and possible input of NDUF-genes expression in CRC progression is still to be established. Overexpression of NDUFA4L2, which is also upregulated in HCT116 oxpl-R cells (log2 fold change = 2.4, data not shown), is reported to be linked with poor prognosis in colon cancer [55] and serves as a prognostic biomarker for most cancers, including colon cancer [56].”
Figure S4: Prognostic value of selected ROS-related genes in colorectal cancer according to the Human Protein Atlas. (a) The boxplots displaying the distribution of normalized counts for selected redox-related genes in HCT116 (sensitive) and HCT116 oxpl-R (resistant) cells, with each point representing normalized count for an individual sample. The significance shown on the plots is based on the adjusted p-values (p.adj) calculated using the Wald test (DESeq2). (b) The Kaplan-Meier curves (survival analysis) assessing the correlation between gene expression and survival for colorectal cancer patients (plots downloaded from the Human Protein Atlas, https://www.proteinatlas.org/).
We believe these revisions have strengthened our manuscript and hope that the improvements meet your expectations. We remain open to any further suggestions and would be happy to address additional requests promptly.
Reviewer 2 Report
Comments and Suggestions for Authors
- Chromosomal alterations (e.g., t(5;6), chr10q deletion) and DEGs (e.g., AKR1C2/3, AHR) lack CRISPR/knockout validation to prove causality in resistance.
- Findings rely solely on in vitro HCT116 cells without validation in patient-derived samples or in vivo models.
- How are Polyploidy-induced gene expression changes distinguished from aneuploidy-specific effects (e.g., chr3/22 amplifications).
- Elevated ROS in resistant cells is observed, but its role (cause vs. consequence of resistance) remains unclear without pharmacological modulation experiments, why?
- Are RNA-seq data (GSE pending) and methodological details (e.g., DESeq2 parameters) insufficient for full reproducibility? Why?
Author Response
Response to Reviewer 2:
We sincerely thank the reviewer for their interest in our manuscript and for the insightful comments. Below, we will address your questions point by point:
- “Chromosomal alterations (e.g., t(5;6), chr10q deletion) and DEGs (e.g., AKR1C2/3, AHR) lack CRISPR/knockout validation to prove causality in resistance.”
- We are grateful to the reviewer for this reasonable remark on the need to confirm chromosomal rearrangements presented in this Article in the context of drug resistance. Though, this is a big task that requires significant time and resources. We intend to perform additional experiments and present the additional validation confirming the role of these rearrangements in resistant cells in a separate publication. This Article is conceptually intended by us as the first representation and general characteristic of oxaliplatin-resistant cells. It will be followed by a series of papers presenting experimental results obtained based on the described cells, with this article serving as the foundational groundwork. - “Findings rely solely on in vitro HCT116 cells without validation in patient-derived samples or in vivo models.”
- We acknowledge that moving to patient-derived data and performing in vivo validation are essential steps to verify our observations. Therefore, we have supplemented the discussion with expression data of specific markers in colorectal cancer patients from an open database. We used the Human Genome Atlas and literature data to provide additional information regarding the prognostic value of reported targets in CRC. As a result of this analysis, a new figure (Figure S4) has been added and discussed: Line 537: “Some of the detected ROS-related genes are reported as prognostic markers for CRC according to the Human protein atlas [52,53]. Changes in gene expression observed in oxaliplatin-resistant HCT116 oxpl-R cells have been associated with CRC patient survival outcomes. Notably, the suppression of the antioxidant gene GPX3 and the upregulation of NCF2 are both associated with poor prognosis (Figure S4). In contrast, NDUFB5, the gene encoding a subunit of NADH dehydrogenase (ubiquinone) and upregulated in resistant cells, is reported to be linked with positive prognosis in CRC according to the survival analysis data. However, active mitochondrial metabolism and biogenesis is reported to be associated with resistant CRC phenotypes [54] and possible input of NDUF-genes expression in CRC progression is still to be established. Overexpression of NDUFA4L2, which is also upregulated in HCT116 oxpl-R cells (log2 fold change = 2.4, data not shown), is reported to be linked with poor prognosis in colon cancer [55] and serves as a prognostic biomarker for most cancers, including colon cancer [56].”. As for in vivo studies, this particular issue lies beyond the scope of the present study. We plan to address it in our subsequent investigations. - “How are Polyploidy-induced gene expression changes distinguished from aneuploidy-specific effects (e.g., chr3/22 amplifications).”
- As we mention in the manuscript, observed expression changes are challenging to interpret and link with particular chromosomal alterations. Line 330: “After tetraploidization, proportions of gene transcripts are supposed to stay the same, however, it’s not just proportional scaling [15,16], so tetra- and octaploids are known to have changed gene expression pattern [17]. While these tetraploidization-induced transcriptomic changes are not directly evident in the karyoplot, the HCT116 oxpl-R genome shows multiple aneuploidies, including three copies of chromosomes 2, 4, 5, 6, 8, 13, 15, and 16, and five copies of chromosomes 3 and 22. These disproportionate karyotype changes are reflected in the transcriptomic data, with specific chromosomes showing enrichment of either down-regulated or up-regulated DEGs.” So, since polyploidy-induced gene expression changes can be disproportional too, all we can do at this point is just match DEGs with their location on the chromosome to make assumptions. For example, DEGs located on chromosome 3 are rather aneuploidy-related. For sure, this approach is straightforward since upstream regulation should be also considered. We have clarified this in text to make this point more obvious. Line 343: “It should be noted that this approach to interpreting the transcriptomic effects of chromosomal rearrangements is somewhat straightforward and simplistic, as it does not take into account possible upstream regulation of gene transcription.” - “Elevated ROS in resistant cells is observed, but its role (cause vs. consequence of resistance) remains unclear without pharmacological modulation experiments, why?”
- We appreciate the reviewer’s insightful comment regarding the role of ROS. We agree that this is an important task. However, a thorough experimental investigation of ROS causality would significantly expand the scope of our current work, which focuses on the general differences between sensitive and resistant cells. We believe that further exploration of ROS in platinum resistance is the subject of a separate study.
To directly address the reviewer's question within the current manuscript, we have added a paragraph (lines [501-507]) discussing the causality of ROS according to literature data: “The question of whether ROS are the cause or the consequence of drug resistance in cancer cells has a dual answer. A body of evidence indicates that ROS are important for both inducing and maintaining drug resistance. ROS can induce drug resistance by promoting genomic instability through DNA damage [40] and activating pro-survival signals that block apoptosis [14,41]. Conversely, maintaining the resistant phenotype requires a critical balance between ROS and antioxidant enzymes, highlighting ROS modulation as a potential therapeutic approach [42,43].”
- “Are RNA-seq data (GSE pending) and methodological details (e.g., DESeq2 parameters) insufficient for full reproducibility? Why?”
- We have aimed to include all the necessary methodological information to enable the reproduction of the analysis. Following your comment, we carefully reviewed the materials and methods section and supplemented it with additional details. DESeq function was used with default parameters, parameters of gseGO and gseKEGG were specified Line 666: ‘GseGO function used with the following parameters: minGSSize = 3, maxGSSize = 800, pvalueCutoff = 0.05, OrgDb = org.Hs.eg.db, the results obtained were simplified using “simplify” method from the package clusterProfiler (cutoff = 0.7, by = "p.adjust", select_fun = min) [62] to reduce redundancy among enriched Gene Ontology (GO) terms; GseKEGG function used with default parameters.’
We believe these revisions have strengthened our manuscript and hope that the improvements meet your expectations. We remain open to any further suggestions and would be happy to address additional requests promptly.
Reviewer 3 Report
Comments and Suggestions for Authors
The authors investigate mechanisms of oxaliplatin resistance in HCT116 cell line using cytogenetic karyotyping and transcriptomic profiling. They report tetraploidization and extensive genomic rearrangements in resistant cells, accompanied by more than 1,800 differentially expressed genes. Enrichment analysis highlights changes in redox balance and metabolic adaptation, with resistant cells showing elevated ROS production and enhanced energy metabolism.
The study proposes a link between structural genomic alterations, transcriptional rewiring, and functional phenotypes, suggesting potential metabolic vulnerabilities as therapeutic targets. Overall, the manuscript provides useful and interesting insights into mechanisms of drug resistance and is supported by an extensive bibliography, demonstrating a thorough review of the relevant literature.
However, several points require improvement to enhance clarity, reproducibility, and adherence to standard scientific presentation:
The title should follow standard capitalization rules: the first letter of each main word should be in uppercase. For example, instead of “Platinum-based chemotherapy resistance remains a critical barrier in colorectal cancer treatment”, it should read “Platinum-Based Chemotherapy Resistance Remains a Critical Barrier in Colorectal Cancer Treatment”.
Line 71: The sentence “The oxaliplatin concentration of 150 μM was chosen based on the results of the MTT test, as the concentration at which 50% of the cells died 48 h after changing the medium to fresh without the active drug (IC50 48h)” appears redundant, as the same information is already stated earlier (line 63). I recommend removing this repetition for clarity and conciseness.
Line 74: The manuscript describes successive rounds of oxaliplatin treatment applied to HCT116 cells; however, the conditions of the control group (HCT116 maintained through successive rounds without treatment) are not specified. In addition, the number of passages the treated cells underwent is not clearly reported, either in the Results or in the Materials and Methods section. Since extensive passaging itself can contribute to genetic alterations, it is important to clarify whether a control group was maintained under the same culture and passage conditions. If not, this should be explicitly acknowledged in the Discussion, as it could represent an alternative explanation for some of the observed alterations.
Line 88: The description of the resistance index (RI) results is detailed; however, the text does not reference the corresponding figure (Figure 1c) where these data are shown. I recommend adding this citation to improve clarity and guide the reader.
Line 93-95. The statement describing passive and active mechanisms of drug resistance is more suitable for the Discussion section rather than the Results, as it provides general background rather than new findings. Additionally, it would be helpful to support this statement with appropriate references to provide contex.
Figure 1D; Although the reagent is mentioned in the Materials and Methods section, it should also be specified in Figure 1D, both in the figure legend and on the graph itself, to ensure clarity and facilitate understanding for readers.
Please ensure that the unit for micromolar is correctly indicated in the graphs (µM) rather than using “uM,” to adhere to standard scientific notation.
I am having difficulty understanding Figure 1B. If I understand correctly, the cells undergo successive rounds of treatment with 150 µM oxaliplatin, so it is unclear why the x-axis of Figure 1B ranges from 0 to 100 µM. Clarification of this point would help the reader interpret the data correctly.
In Figure 1D, it is unclear why 25 µM oxaliplatin was used for the experiment. Please clarify the rationale for choosing this concentration.
In the graph of Figure 1D, it is unclear what the value “24” on the x-axis represents. Please clarify this labeling to avoid confusion for the reader.
Line 113 and Figure 1E: The manuscript lists the concentrations used for oxaliplatin (10 µM), cisplatin (25 µM), etoposide (30 µM), and doxorubicin (0.5 µg/mL), but it is unclear how these concentrations were determined. Please provide the rationale or references for selecting these specific doses.
In Figure 3d, if the genes shown are human, their names should be formatted in uppercase and italics, in accordance with standard gene nomenclature.
It is recommended to mention the figure in the text before presenting it. This helps guide the reader, providing context and making it clear what to focus on. Placing the figure before it is referenced can cause confusion and disrupt the logical flow of the manuscript.
Please consider avoiding the frequent use of the first person (“we”) throughout the manuscript. Using a more neutral or passive voice can improve readability and maintain a formal scientific tone.
Line 538: the temperature unit is missing the degree symbol. It should read “37 °C” instead of “37С” to correctly indicate degrees Celsius.
Line 619, it would be clearer to present the primers used in a table rather than in the text. This improves readability and allows the reader to quickly reference sequences.
In the Results section, it is recommended to focus on presenting the experimental findings without including interpretations or speculations. Any discussion or explanation of the results should be reserved for the Discussion section and properly referenced to maintain clarity and adhere to scientific standards.
I could not identify a separate Conclusions section in the manuscript. It is recommended to include a distinct Conclusions section rather than embedding conclusions within the Results, as this improves clarity and helps the reader clearly understand the key findings and implications of the study.
Author Response
Response to Reviewer 3:
We appreciate the reviewer’s careful reading of our work and the valuable feedback provided. Your suggestions have significantly strengthened the manuscript and enhanced its overall quality and clarity. We are especially grateful for the minor yet valuable comments related to the presentation and enhancing the text’s clarity for the reader, we appreciate your input. Below, we will address your questions point by point:
- “The title should follow standard capitalization rules: the first letter of each main word should be in uppercase.”
- Thank you for pointing that out. The title of the manuscript was corrected according to the rules. - “Line 71: The sentence “The oxaliplatin concentration of 150 μM was chosen based on the results of the MTT test, as the concentration at which 50% of the cells died 48 h after changing the medium to fresh without the active drug (IC50 48h)” appears redundant, as the same information is already stated earlier (line 63). I recommend removing this repetition for clarity and conciseness.”
- You are right, this section includes repeats. We have revised the whole paragraph. - “Line 74: The manuscript describes successive rounds of oxaliplatin treatment applied to HCT116 cells; however, the conditions of the control group (HCT116 maintained through successive rounds without treatment) are not specified.” “Since extensive passaging itself can contribute to genetic alterations, it is important to clarify whether a control group was maintained under the same culture and passage conditions. If not, this should be explicitly acknowledged in the Discussion, as it could represent an alternative explanation for some of the observed alterations.”
- Indeed, prolonged culturing itself can lead to genetic rearrangements and the development of resistance, but in this case, the main focus is the resistance itself rather than the mechanism of its formation. Moreover, we observe selective resistance to platinum drugs (cross-resistance) and only weak cross-resistance to other drugs (Figure 1e), which supports the conclusion that platinum exposure, rather than the duration of culturing, is the key factor driving resistance development. It would be interesting to see whether the control HCT116 cell lines demonstrate a comparable level of drug resistance after prolonged culturing, however, this cannot be examined within the scope of the current review. Thank you for this remark, these reasonings were added to the Discussion. Line 391: “The origin of the platinum resistance observed in HCT116 oxpl-R cells results from multiple cycles of high-concentration platinum drug exposure. Prolonged culturing itself is known to enable cells to develop resistance to treatment [50,51], and in this case, the cells went through 26-28 passages during the acquisition of oxaliplatin resistance. According to our results, oxaliplatin-resistant cells HCT116 oxpl-R demonstrate selective resistance to platinum drugs (cross-resistance) and only a slight cross-resistance to other DNA-damaging drugs (Figure 1e), supporting the conclusion that platinum exposure, rather than the duration of culturing, is the key factor driving resistance development in these cells.”
“In addition, the number of passages the treated cells underwent is not clearly reported, either in the Results or in the Materials and Methods section.”
- At the time of cryopreservation, the HCT116 oxpl-R cells had undergone 26-28 passages. We added this information to the Materials and Methods section. - “Line 88: The description of the resistance index (RI) results is detailed; however, the text does not reference the corresponding figure (Figure 1c) where these data are shown. I recommend adding this citation to improve clarity and guide the reader.”
- The reference was added, thank you. - “Line 93-95. The statement describing passive and active mechanisms of drug resistance is more suitable for the Discussion section rather than the Results, as it provides general background rather than new findings. Additionally, it would be helpful to support this statement with appropriate references to provide context."
- Line 100: “Drug resistance of tumor cells can be either passive due to a decrease in the intracellular concentration of the active substance, or active, which involves the restoration of damage and survival under conditions of cellular stress caused by the action of chemotherapeutic drugs. In order to identify the type of resistance formed, …” - This statement is not a piece of the discussion but rather a short explanatory note, which is important to justify the experiment described in this section. Certainly, you are right that such statements require literature support, and we have added the appropriate references to the text. - “Figure 1D; Although the reagent is mentioned in the Materials and Methods section, it should also be specified in Figure 1D, both in the figure legend and on the graph itself, to ensure clarity and facilitate understanding for readers.”
- As recommended by the reviewer, the compound used (oxpl) is now presented in the Y-axis labels of the corresponding graphs in Figure 1d. - “Please ensure that the unit for micromolar is correctly indicated in the graphs (µM) rather than using “uM,” to adhere to standard scientific notation.”
- All instances of uM in figures were changed to µM. - “I am having difficulty understanding Figure 1B. If I understand correctly, the cells undergo successive rounds of treatment with 150 µM oxaliplatin, so it is unclear why the x-axis of Figure 1B ranges from 0 to 100 µM. Clarification of this point would help the reader interpret the data correctly.”
- The reviewer is correct that the selection process employed 150 µM oxaliplatin. However, Figure 1b characterizes the resulting resistant phenotypes, not the selection process itself. The dose-response curves in Figure 1b were generated by treating the resistant populations from rounds 2-4 with a titration of oxaliplatin (0-100 µM) to precisely measure their acquired IC50 values, which are then plotted in Figure 1c. We have added reference to this plot in text to clarify this. - “In Figure 1D, it is unclear why 25 µM oxaliplatin was used for the experiment. Please clarify the rationale for choosing this concentration.”
- The concentration of 25 µM oxaliplatin was selected based on the dose-response curves (Figure 1b). This concentration induces a moderate toxic effect in sensitive cells while showing only a slight effect in resistant cells, thereby clearly illustrating the differential response between the two cell lines. This rationale was added to the corresponding section: Line 105. - “In the graph of Figure 1D, it is unclear what the value “24” on the x-axis represents. Please clarify this labeling to avoid confusion for the reader.”
- Corrected - “Line 113 and Figure 1E: The manuscript lists the concentrations used for oxaliplatin (10 µM), cisplatin (25 µM), etoposide (30 µM), and doxorubicin (0.5 µg/mL), but it is unclear how these concentrations were determined. Please provide the rationale or references for selecting these specific doses.”
- The drug concentrations used for viability tests (etoposide, doxorubicin) were selected based on the established IC50 values for the parental, platinum-sensitive HCT116 cell line. These IC50 values were determined in our preliminary dose-response experiments (data not shown). Using concentrations near the IC50 for the sensitive cells allows for a clear and effective comparison of the differential response between the sensitive (parental) and resistant cell populations. We have now added this explanation to the Results section (Line 126) to clarify the rationale for the chosen doses - “In Figure 3d, if the genes shown are human, their names should be formatted in uppercase and italics, in accordance with standard gene nomenclature.”
- In the general practice of DESeq2 visualizations, gene names in heatmaps and volcano plots are by default displayed in uppercase letters without italics. This formatting aligns with common conventions for gene symbols in relevant databases (e.g., HGNC), which are typically presented in all caps and non-italicized to ensure clarity and avoid confusion with other scientific notation. The default non-italicized uppercase style is widely accepted in bioinformatics visualizations and publications and is consistent with common practice in DESeq2 and related visualization tools. - “It is recommended to mention the figure in the text before presenting it.”
- Thank you for the fair remark. We have moved Figure 4 further down in the text so that it does not precede its reference. - “Please consider avoiding the frequent use of the first person (“we”) throughout the manuscript. Using a more neutral or passive voice can improve readability and maintain a formal scientific tone.”
- We agree with this statement and replaced “we” in the manuscript text with passive voice wherever possible. - “Line 538: the temperature unit is missing the degree symbol. It should read “37 °C” instead of “37С” to correctly indicate degrees Celsius.”
- Corrected - “Line 619, it would be clearer to present the primers used in a table rather than in the text. This improves readability and allows the reader to quickly reference sequences.”
- We thank the reviewer for this recommendation. We have organized our primer data in a table (Table 2). - “In the Results section, it is recommended to focus on presenting the experimental findings without including interpretations or speculations. Any discussion or explanation of the results should be reserved for the Discussion section and properly referenced to maintain clarity and adhere to scientific standards.”
- We agree with the reviewer's recommendation to maintain a clear distinction between results and their interpretation. We have retained only the minimal explanations necessary to justify a specific methodological choice or to provide logical continuity for the reader. - “I could not identify a separate Conclusions section in the manuscript.”
- According to IJMS guidelines, the Conclusions section is mandatory and recommended for complex articles. We considered it redundant in this case.
We believe these revisions have strengthened our manuscript and hope that the improvements meet your expectations. We remain open to any further suggestions and would be happy to address additional requests promptly.
Round 2
Reviewer 3 Report
Comments and Suggestions for Authors
The authors have addressed the majority of my previous suggestions point by point, which is appreciated. However, I recommend the following additional points:
Comment 1: The GenBank accession numbers for the genes studied should be added to the primer table. This information is essential for accurately identifying the gene sequences used and for ensuring reproducibility of the experiments.
Comment 2: I do not agree with omitting a Conclusions section. While the authors may consider it redundant, it is important to recognize that some readers focus only on specific sections of a manuscript rather than reading it in its entirety. Including a Conclusions section would improve clarity and accessibility of the key findings.
Author Response
We would like to thank the reviewer for the prompt revision. Both of your comments have been taken into account.
Comment 1: The GenBank accession numbers for the genes studied should be added to the primer table. This information is essential for accurately identifying the gene sequences used and for ensuring reproducibility of the experiments.
We appreciate the reviewer's suggestion to include GenBank accession numbers for the genes studied. However, we would like to clarify our primer design strategy. The primers were designed to target all known coding transcript variants of each gene, rather than a single specific splice variant. This approach ensures that the PCR product is independent of alternative splicing events and represents the gene comprehensively.
Given this, specifying a single NCBI RefSeq accession number is limiting because it corresponds to a particular transcript variant. Instead, our primers cover conserved regions across all protein-coding transcripts, maximizing detection of gene expression regardless of splice isoforms. This strategy improves robustness and reproducibility in expression analysis where multiple transcript variants exist.
To make our primer design strategy transparent, we have added this explanation to the corresponding section of material and methods and provided gene identifiers (Gene IDs) in the primer table for clarity and reproducibility.
Comment 2: I do not agree with omitting a Conclusions section. While the authors may consider it redundant, it is important to recognize that some readers focus only on specific sections of a manuscript rather than reading it in its entirety. Including a Conclusions section would improve clarity and accessibility of the key findings.
We agree that readers often rely on the Conclusions section to understand the core message of the paper. We have included this section to briefly summarize our findings and conclusions.
Thank you for your attention to detail and valuable comments. We are open to any further suggestions.